# Challenges and Opportunities for Extracellular Vesicles in Clinical Oncology Therapy

**DOI:** 10.3390/bioengineering10030325

**Published:** 2023-03-03

**Authors:** Shuya Lu, Qingfa Cui, Huan Zheng, Yuan Ma, Yanchun Kang, Ke Tang

**Affiliations:** 1Department of Biochemistry and Molecular Biology, Tongji Medical College, Huazhong University of Science and Technology, Wuhan 430030, China; 2Cell Architecture Research Center, Huazhong University of Science and Technology, Wuhan 430030, China

**Keywords:** extracellular vesicles, microparticles, drug delivery, clinical therapy, tumor immunotherapy

## Abstract

Extracellular vesicles (EVs) are membrane-bound vesicles that can be released by all cell types. They may have different biogenesis, physical features, and cargo. EVs are important biomarkers for the diagnosis and prediction of many diseases due to their essential role in intercellular communication, their highly variable cargoes, and their accumulation in various body fluids. These natural particles have been investigated as potential therapeutic materials for many diseases. In our previous studies, the clinical usage of tumor-cell-derived microparticles (T-MPs) as a novel medication delivery system was examined. This review summarizes the clinical translation of EVs and related clinical trials, aiming to provide suggestions for safer and more effective oncology therapeutic systems, particularly in biotherapeutic and immunotherapeutic systems.

## 1. Introduction

The secretory process of EVs has been highly conserved throughout evolution; almost all prokaryotic and mammalian cells can release vesicles into the extracellular environment for message exchange [1,2,3,4]. EVs can be divided into the following subgroups. (1) Exosomes, released after the fusion of intracellular vesicles and the plasma membrane, with diameters of 30–150 nm. (2) Ectosomes or shedding microvesicles (as known as microparticles, MPs), with diameters of 100–1000 nm. These cells protrude outward through the envelope of the mother cell to wrap intracellular material and release it. Their surface is a phosphorous structure that expresses the special antigen expressed by the parental cell. (3) Migrasomes, with a diameter of 500–3000 nm. (4) Large oncosomes, with a diameter of 1000–10,000 nm and an apoptotic body with a diameter of 50 nm–2 μm [5]. Additionally, a new type of EV called exomeres, which are described as tiny non-membranous particles with a diameter of less than 50 nm, was recently discovered by David Lyden and colleagues. However, the biogenesis and secretion processes of exomeres remain unclear, and even the isolation methods for exomeres are different in different studies [6,7].

In recent years, EVs have been found in a variety of physiological fluids including blood, urine, saliva, breast milk, amniotic fluid, ascites, cerebrospinal fluid, bile, and semen [8,9,10,11,12]. More importantly, EVs can be directly isolated from physiological fluid in solid organs including the liver, heart, brain, and others [13,14,15,16,17]. Originally, the majority of studies focused on clinical diagnosis; however, in recent years, EVs have shown a bright future as an excellent therapeutic vehicle for treating diseases, especially cancer. Many reviews have reported the use of EVs in clinical trials on tumors, where they induced immune responses. In this article, we review the use of MP effectors in clinical applications in oncology.

Due to their unique biocompatibility and size, EVs initially showed considerable potential as drug carriers and hold a place in cancer therapeutic interventions. Recently, we explored the role of drug-loaded EVs (also known as drug-loaded microparticles) in the activation of the immune system, suggesting the possibility of immunotherapy for the treatment of malignancies [18]. During the formation of MPs, not only are the cell membrane components remodeled, but also the cytoskeleton. This remodeling leads to the disappearance of the local centripetal force, while the corresponding reaction force still exists, causing the cell membrane to bulge outward. During apoptosis, the apoptotic signal activates the small G protein Rho-associated coiled-coil containing kinases I (ROCKI) through the caspase pathway, and the activated ROCKI activates the motor protein by phosphorylating the myosin light chain, thereby promoting cytoskeleton actin microfilament remodeling and the release of apoptotic MPs [19]. It is unclear whether cytoskeletal remodeling is coupled with membrane remodeling and whether there is an interaction. Here, we summarize the clinical trials and translation applications of microparticles in cancer immunotherapy to help researchers better understand this field and to promote the application of microparticles.

## 2. EV Isolation Methods

According to the Minimal Information for Studies of Extracellular Vesicles 2018 (MISEV2018, https://onlinelibrary.wiley.com/doi/10.1080/20013078.2018.1535750, access date: 10 November 2022), separating EVs completely from other components or obtaining completely pure EVs is basically impossible to achieve using current technology [20]. MISEV’s global survey showed that differential ultracentrifugation was the most commonly used method for isolating EVs at the end of 2015; it is also the technique used in our laboratory [21]. Other techniques such as density gradients, size exclusion chromatography, and immunocapture were used in about 5–20% of cases. Next, we briefly describe the methods commonly used to isolate EVs in a centralized manner.

### 2.1. Differential Centrifugation

The original method of isolating EVs from the above-mentioned biological fluids is ultracentrifugation, also known as ‘density gradient centrifugation’. This method is currently considered to be the ‘gold standard’ [22]. Nevertheless, differential centrifugation frequently produces disappointing outcomes such as relatively low yields or insufficient purity of the isolated EVs (Table 1). Furthermore, researchers often use the same differential centrifugation techniques on numerous rotors while omitting variations in sample viscosity. Due to the time-consuming nature of these differential centrifugation procedures and the large number of mother cells required for EV separation, they frequently produce dubious results.

Based on sample density, density gradient centrifugation is a more exacting ultracentrifugation technique that is more effective in avoiding contamination from protein co-precipitation and the further purification of vesicles. To introduce this particular separation technique, we used the example of iodixanol in this instance [23]. EVs were resuspended in Tris buffer containing 30% (wt/vol) iodixanol, followed by sequential addition of 20% (wt/vol) iodixanol and 10% (wt/vol) iodixanol [24]; the solution was then ultracentrifuged. The lipid-rich and membrane-encapsulated EVs floated upward, which did not occur with protein aggregates. In this manner, the purification of EVs was enhanced. It is important to note that several EVs are separated simultaneously, as different molecules settle at different rates during ultracentrifugation. Iodixanol density gradient centrifugation (equivalent to ultracentrifugation) is more effective than sucrose in maintaining the biophysical properties of vesicles, but it may affect the biological function of EVs (hypertonicity of the gradient solution leads to subcellular water loss)

### 2.2. Ultrafiltration

Ultrafiltration is one of the most common size-based techniques used for EV isolation [5]. EVs larger than a given filter are retained by the filter, while smaller EVs enter the filtrate. This method is relatively simple and efficient, and does not affect the biological activity of the EVs, but its disadvantage is that EVs may clog the filter pores, leading to shorter membrane lifetimes and lower separation efficiency. In addition, the EVs trapped on the membrane may become adherent, resulting in reduced access to exosomes [5,25].

### 2.3. Size-Exclusion Chromatography

Size-exclusion chromatography (SEC) is a simple, inexpensive, and reliable method for the size-based separation of EVs [26]. This method allows EVs of different sizes to pass through gels with pores of specific sizes to separate EVs [27]. When the sample passes through the gel (mainly hydrophilic gels, polystyrene gels, and inorganic fillers), larger EVs do not penetrate the pores and are directly eliminated, whereas smaller EVs can completely penetrate the pores before being eliminated. This method allows for the separation of EVs from other soluble proteins, thus improving the purification of the isolated EVs [28]. SEC does not require the separation of EVs at high speed, which yields purer EVs with their biological activity preserved [29]. When separating different samples, the columns need to be cleaned and re-equilibrated, which may result in longer separation times.

### 2.4. Immunoaffinity Capture

Immunoaffinity capture-based techniques rely on an antibody based on the antigens expressed on the exosome’s surface to capture EVs [30]. Antibodies against specific antigens of interest can be attached to plates, magnetic beads, resins, and microfluidic devices [31]. Thus, only EVs positive for a particular antigen are captured, while negative EVs are lost. A significant benefit of these techniques over others is that they allow for the isolation of EVs derived from a specific source. Immunoaffinity methods not only have the potential to help isolate specific EV subsets from complex mixtures, but may also distinguish certain EVs from other EVs, if EV-specific markers are identified [32]. Fractions of EVs that do not express any antigens are excluded. Immunoaffinity methods require explicit knowledge of specific markers for each EV class. However, this knowledge is not widely available yet. Thus, this purification method will cause results in mixed populations of different EVs. Considering the complexity of biological fluids such as plasma, immunoaffinity capture-based techniques are usually used after exosome enrichment by ultracentrifugation or ultrafiltration.

This method is suitable for the isolation of highly pure exosomes from a specific source with minimal influence on the structure and morphology of the exosomes [33]. Therefore, this method is applied to enrich and characterize unique exosomes. However, the cost of immunoaffinity capture methods is considerably high. In addition, pH changes and the addition of salt reagents during exosome elution may also affect subsequent studies and applications.

### 2.5. Microfluidic Filtration

Microfluidic filtration refers to the separation of liquids containing EVs by different methods after their addition to the sheath medium, depending on their physical and biochemical properties. Microfluidic filtration can separate EVs based on a variety of different properties such as size, density, and electric charge. Various microfluidic filtration systems have been developed, most of which use size to separate EVs [34]. Similar to the SEC, microfluidic filtration is simple and reliable. Microfluidic filtration can obtain high-purity EVs with a single characteristic in large quantities [35]. However, due to shear stress, the deformation of EVs may occur during the microfluidic filtration. Since the separation is based on a single characteristic, the infiltrated EVs will lose their heterogeneity to a certain extent.

### 2.6. Other Separation Methods

In 2022, Chen Gang et al. developed a non-contact “negative separation” strategy [36]. They used separation materials that directly bind to non-tumor-cell-derived EVs to eliminate non-tumor-cell-derived EVs, thus enabling the specific collection of native tumor EVs (T-EVs) from the tumor tissues. By adding antibody-conjugated magnetic microparticles (MMs) to heterogeneous EVs extracted from tumor tissues, non-tumor cell-secreted EVs were removed by magnetic separation [2]. In this way, selective non-destructive T-EVs were acquired. Compared with the traditional immunomagnetic sphere-based strategy, the isolated material does not need to bind to the target tumor extracellular vesicles, and thus ultimately preserves the natural properties and biological functions of T-EVs [37].

All methods used to isolate EVs have their advantages and disadvantages; thus, choosing the most appropriate method for a specific application is not an uncomplicated task. This issue is very important because the isolation technique may affect the molecular content and biological activity of the isolated EVs. Herein, the establishment of purification standards is essential to obtain reproducible data among all research laboratories and to facilitate EV adoption in clinical settings.

The first question that needs to be answered regarding the clinical translational application of exosomes in cancer therapy is how to produce clinical-grade GMP (good manufacturing practice) exosomes [38]. We hold the view that the traditional ultracentrifugation method has the advantages of simplicity of operation and low cost. Codiak adopts a GMP standard bioreactor-based method for large-scale clinical-grade exosome production. The standard operating procedure produces clinical-grade GMP-engineered exosomes (iExosomes) with oncogenic capacity [39]. Briefly, the production of GMP iExosomes uses bioreactors to mass culture cells. The bioreactor allows for the periodic collection of large volumes of culture supernatants, each of which is sterile, endotoxin-free, and mycoplasma-free.

**Table 1 bioengineering-10-00325-t001:** Summary of the isolation methods for EVs.

Isolation Strategy	Separation Basis	Principle	Reference
Density gradient centrifugation	Density	EVs are resuspended in Tris buffer containing sucrose gradient and ultracentrifuged; then, EVs stay in the medium layer with similar density.	[23,24]
Ultrafiltration	Size	A filter retains larger EVs while allowing smaller EVs to pass.	[5,25]
Size-exclusion chromatography	Size	EVs pass through materials with pores of specific sizes; larger ones excluded and eliminated while smaller ones penetrate the pores.	[26,27,28]
Immunoaffinity capture	Antigen on the exosome’s surface	Antibodies to specific antigens are attached to resins, with antigen-positive EVs captured and antigen-negative EVs lost.	[30,32]
Microfluidic filtration	Various properties	Based on size, density, electric charge, etc.	[2,36]
Negative separation	Antigen on the exosome’s surface	Separation materials are bound to non-tumor-cell-derived EVs to remove them through MMs.	[34,40]

MMs, magnetic microparticles.

Almost all eukaryotic cells including erythrocytes and platelets, are capable of releasing MPs. The mechanisms of MP production and release and their biological functions have been intensively studied [41,42]. MP release from nucleated cells usually occurs during activation or early apoptosis as well as due to cell differentiation, stress, senescence, stimulation by external cytokines or shear forces, ATP (adenosine triphosphate) treatment, apoptosis, microenvironmental changes, hypoxia, and malignancy. Thus, MPs appear to function as carriers to pass information molecules between cells. Considering the similarities in size, structure, and carrier function between MPs and nanoparticles, it is reasonable to assume that MPs can act as endogenous natural carriers for the delivery of chemotherapeutic drugs.

## 3. Potential of MPs in Delivering of Chemotherapy Drugs and Other Cargos

Chemotherapy remains one of the effective approaches for treating clinical tumors; however, precise delivery of chemotherapy drugs to tumor sites with low toxicity is still a challenge. In recent decades, several nanocarriers for loading chemotherapy drugs such as liposomes and albumin have been approved for clinical use. However, these synthetic nanomedicines do not meet the clinical needs for two main reasons: first, they cannot overcome the chemo-resistance of tumor cells; second, they are poorly biocompatible, leading to the long-term accumulation of toxicity in vivo. However, is it possible that a new, low-toxicity chemotherapeutic drug carrier from nature already exists?

In 2012, we established a new natural bio-derived nanodrug delivery system in which chemotherapeutic drugs are encapsulated by tumor-cell-derived MPs [18]. This system has the advantages of efficient targeted delivery, anti-tumor activity, and low toxicity. After more than 10 years of basic and clinical translational research, we discovered that it plays a crucial role in activating innate and adaptive immunity, in addition to its direct drug delivery function, showing a wide therapeutic potential in the clinical treatment of various tumors [43,44].

### 3.1. Tumor Microparticles (T-Mps) Are Biological Drug Carriers

Chemotherapeutic agents (methotrexate, cisplatin, paclitaxel, etc.) are co-incubated with tumor cells and then irradiated using ultraviolet (UV) light to damage the cytoskeleton and trigger apoptosis, resulting in the release of drug-loaded T-MPs from the cells (Figure 1A) [18,45]. Subsequently, drug-loaded MPs were used to treat a tumor-bearing mouse model; tumor growth was found to be inhibited and mice survived longer, which indicated that T-MPs were effective in killing tumor cells in vitro and in vivo. Meanwhile, mice in the T-MP group did not exhibit significant side effects compared to the chemical drugs [18]. Next, we found that drug-loaded MPs could be efficiently enriched in tumor regions and efficiently deliver chemotherapeutic drugs into intracellular space. Drug-loaded MPs can be released again after triggering the apoptosis of tumor cells, producing a “domino-like” effect. In addition, paclitaxel-loaded MPs can provide greater antitumor effects with the same drug content as clinically licensed liposomes of lipusu (paclitaxel liposomes). While providing the same anticancer effect, the liposomal drug exhibited significant chemotherapeutic toxicity, while the drug-loaded MPs showed a favorable safety profile. In a human ovarian nude mouse tumor model, we also tested two types of drug-loaded MPs (paclitaxel and cisplatin), showing that combinations of different drug-loaded MPs had more potent anti-tumor functions and even completely cured the tumors in some mice.

### 3.2. T-MPs Reverse Drug Resistance in Tumor-Repopulating Cells

Previous research has shown that tumor heterogeneity is a crucial factor in tumor resistance and metastasis [46]. It is well-known that a subpopulation of tumor stem cells with the ability to self-reproliferate is responsible for tumor drug resistance. Cell softness is a kind of biomechanical property; tumor tissue is usually stiffer than normal tissue due to abnormal production and the crosslinking of extracellular matrix proteins, but individual tumor cells are usually softer than non-malignant cells. Current research has proved that the reduction in tumor cell stiffness is related to the transformation, malignancy, and metastasis of cancer cells [47,48]. Therefore, we established a method based on the biomechanical screening and three-dimensional soft fibrin gel culture of functional cancer stem cells (defined as “tumor-repopulating cells”, TRC) [49].

In our study, we found that TRCs can take up MPs more efficiently than fully differentiated cells, based on the fact that they have soft characteristics. When Japlakinolide, a macrocyclic peptide that reduces cell toughness by stimulating actin polymerization, enhanced the stiffness of these resistant cells, their drug phagocytosis efficiency was significantly reduced [50]. Since the nuclear transport of these drugs may depend on the centripetal movement of lysosomes and the opening of the nuclear pore complex, we further loaded T-MPs with the auto-fluorescent chemotherapeutic drug doxorubicin and found that the T-MPs were able to efficiently transport doxorubicin with red fluorescence into the nucleus [51]. In addition, we carried out a prospective clinical trial in patients with cisplatin-resistant malignant pleural effusion and non-small cell lung cancer [52]. After thoracic infusion of T-MP loaded with cisplatin, drug-resistant tumor cells were effectively destroyed. Additionally, there was a noticeable decline in the number of TRCs. Clinical experiments have also demonstrated the effectiveness of drug-loaded MPs in reversing TRC drug resistance (Figure 1B).

Non-muscle invasive bladder cancer (NMIBC) is usually treated by surgical resection and local chemotherapy of the bladder. Approximately half of the patients experience severe recurrence after treatment because this approach does not completely eliminate drug-resistant tumor regenerating cells. Therefore, there is an urgent need to create novel therapeutic techniques to improve drug sensitivity and eliminate TRCs. We performed bladder cancer perfusion with MPs loaded with chemotherapeutic agents [53]. We found that pre-infusion of T-MPs improved the inhibitory effect of bladder chemotherapy on orthotopic bladder cancer and significantly increased the blood–urine volume in a mouse model. T-MPs were able to raise the pH of lysosomes from 4.6 to 5.6, facilitating the movement of drug-carrying lysosomes through microtubules to the nucleus and their release when it reaches the nucleus [51]. Based on this information, we found that the GTP-binding protein (guanosine triphosphate-binding protein) that mediates lysosomal transport, the Rab7 protein (Ras-related GTP-binding protein 7), is anchored on the lysosomal membrane after binding to GTP, and through this effector molecule RILP (Rab7-interacting lysosomal protein), the dynein is recruited to the lysosome, and once it reaches the lysosomal membrane, the dynein aggregates to form the Rab7 dynein complex, which acts as a link between the lysosome and microtubules. Lysosomal dynein is introduced into the nucleus along the microtubules, and then the drug is released near the nucleus and enters the nucleus through nuclear pores [54] (Figure 2). Based on this mechanism, T-MPs could be widely used as chemosensitizers to improve the chemotherapeutic effect of NMIBC in this way.

### 3.3. T-MP-Loaded Oncolytic Virus

In addition to chemotherapeutic drugs, we found that T-MPs can also successfully transport biomolecules including oncolytic viruses. When encapsulated by T-MPs, oncolytic viruses stay longer at the tumor site and have a larger viral DNA copy number than oncolytic viruses alone [55]. T-MP encapsulation allows oncolytic viruses to better proliferate, precisely target tumor sites, and enter the nucleus of tumor cells for DNA replication, producing a good oncolytic effect. T-MP-mediated oncolytic viral therapy has a number of benefits over oncolytic virus therapy alone including the following: T-MP-loaded oncolytic virus prevents the host from changing antiviral antibodies and does not require virus-specific receptors to enter tumor cells, as the virus instead enters the nucleus, where it is tumorolytic. T-MPs can efficiently transfer the oncolytic virus to the nucleus of the tumor cells and TRCs during oncolytic virus therapy. Although oncolytic viruses have been used in the past, they still need to be strengthened in order to continue killing tumor cells. Additionally, only particular tumor cells that express oncolytic viral receptors can be infected by oncolytic viruses. They can effectively infiltrate tumor cells after being loaded with T-MPs in a receptor-independent manner. There are many opportunities for clinical applications of this breakthrough oncolytic virus vector system in malignancies (Figure 1C).

**Figure 1 bioengineering-10-00325-f001:**
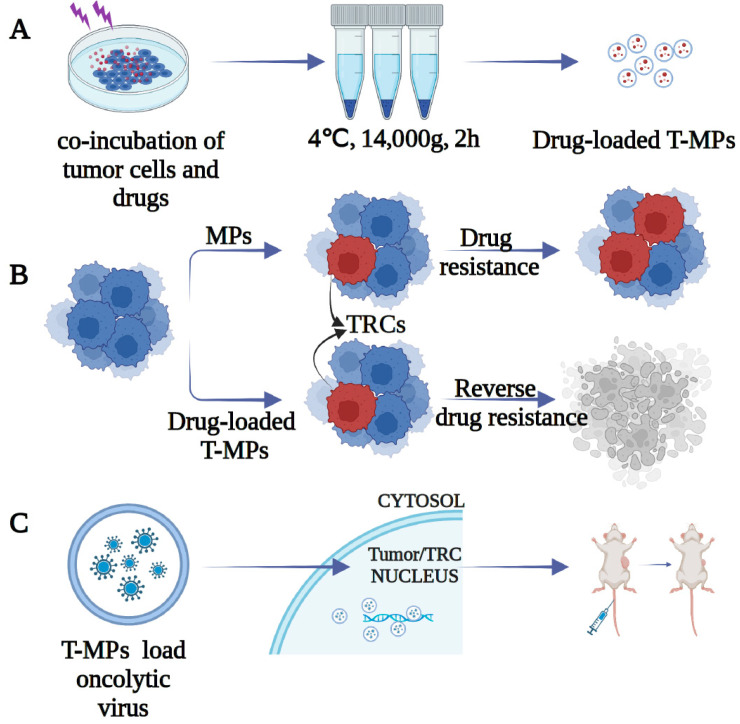
MP potential for delivering chemotherapy drugs and other cargos. (**A**) Chemotherapy-loaded T-MPs [18] (Nature Communication, 2012); (**B**) drug-loaded T-MPs reverse TRC drug resistance [50] (Cell Research, 2016); (**C**) T-MPs carry oncolytic viruses into the nuclei of tumor cells based on [56] (Cancer Immunology Research, 2015).

**Figure 2 bioengineering-10-00325-f002:**
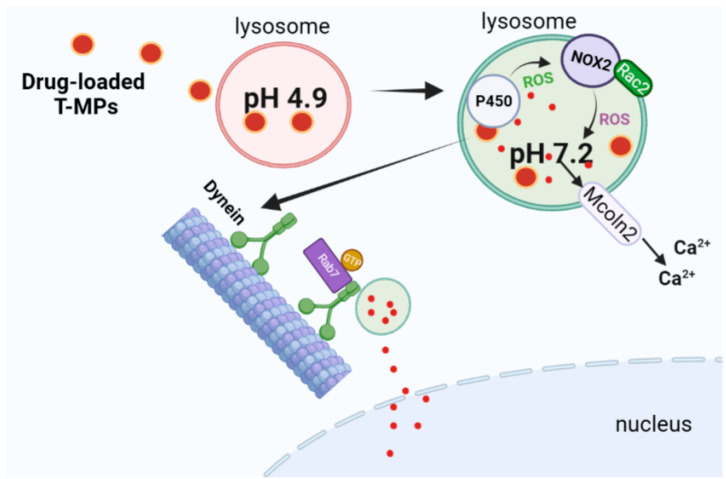
Drug-loaded T-MPs deliver drugs into the nucleus based on [54] (Signal transduction and targeted therapy, 2023).

## 4. Dual Significance of T-MPs in Immune System Regulation

### 4.1. T-MPs Improve Adaptive Immunity

The tumor microenvironment is complex. In addition to the diversity of tumor cells, the immune system is another significant category [57]. The tumor immune microenvironment mediates the phenotypic polarization of the immune cells and stimulates tumor development. Aside from the tumor cells themselves, can immune cells effectively phagocytose T-MP? According to our research, immune cells such as macrophages and dendritic cells can effectively phagocytose T-MPs, while T cells and B cells are less effective [50,58].

Although tumor-associated macrophages (TAMs) have been mentioned as prospective targets for cancer therapy, little is known about how these cells develop and operate in neoplastic settings [59]. As a tumor-specific marker, M2-type TAM promotes tumor immunosuppression, angiogenesis, and tumor growth as well as tumor cell survival, chemotherapeutic treatment resistance, and the formation of TRCs. Since M2-type TAMs promote tumor growth, targeting them is regarded as a promising anticancer treatment. We discovered that the cyclic GMP-AMP synthase (cGAS)/ stimulator of interferon genes (STING)/ Tank binding kinase 1 (TBK1)/ signal transducer and activator of transcription 6 (STAT6) pathway is activated by T-MPs, causing M2-type polarization in macrophages [60]. M2-type macrophages change the pre-metastatic microenvironment by releasing multiple factors to inhibit anti-tumor immune response, promote tumor growth and metastasis, and enhance the development of cancer stem cells. In our study, T-MPs crossed the blood barrier and were educated into M2 macrophages after taken up by tumor resident macrophages[58]. In situ macrophages take up T-MPs and are cultured into M2 macrophages. This study provides fresh insight into the crucial role of T-MPs in altering the tumor microenvironment.

Previously, we demonstrated that T-MPs may deliver chemotherapeutic drugs or oncolytic adenoviruses to the nuclei of TRCs and eradicate them through lysosome-mediated mechanisms [53]. Furthermore, TRCs frequently suppress antitumor immunity to escape the immune system. Can drug-loaded MPs improve the immune responses to tumors by targeting TRCs? Chemotherapy and radiation therapy are the standard treatments used in clinics to kill tumor cells. Both techniques have some effectiveness in anti-tumor therapy, but they also have pronounced drawbacks. Whether it is systemic chemotherapy or local infusion, chemotherapy kills tumor cells, but can also severely harm healthy human cells, leading to toxic and adverse consequences that have a negative impact on efficacy. We demonstrated that low-dose irradiation (LDI) enhanced the effect of cisplatin-coated T-MP (Cis-MP) on TRCs, which inhibited the tumor growth in various tumor types [61]. This antitumor effect killed TRCs and reduced their stimulation of macrophages, which reverses and re-educates tumor-associated macrophages, reversing of the macrophage phenotype from M2 to M1 polarization, in addition to directly killing tumor cells.

Soil is required for tumor growth and metastasis. Several studies have shown that primary tumors can influence the microenvironment of distant organs by secreting a number of cytokines even before metastasis, making it favorable for the metastasis and colonization of tumor cells [62,63,64]. Pre-metastatic niche formation is the term used to describe this process. Studies have revealed that tumor cells release a large number of T-MPs into peripheral circulation when hypoxia (1% O_2_) is induced [65]. These T-MPs then penetrate lung tissue and are taken up by macrophages, triggering the release of the chemokine CCL2. In the study, T-MPs made lung endothelial cells more permeable, which allowed fibrinogen to enter the bloodstream and be deposited as fibrin, leading to increased permeability of the lung endothelium. The survival and growth of TRCs were similarly influenced by these physical microenvironments. According to this study, T-MPs have the ability to change the immune microenvironment of the lung, affecting the extracellular matrix of the lung tissue and creating an inflammatory pre-metastatic milieu that is conducive to tumor cell survival and development. The authors thoroughly examined how T-MPs affect immune cells during the tumor metastatic process.

MP is an important carrier of RNA, but the study of its biological role is still at an early stage. Following the phagocytosis of vesicles, we discovered that many non-coding RNAs are enriched for T-MPs. These non-coding RNAs, in combination with the lysosomal localization characteristics, mediate further lysosome acidification and calcium release, which, in turn, mediates inflammasome NLRP3 (NOD-, LRR- and pyrin domain-containing protein 3) activation, pro-IL-1β splicing, and mature form release [66]. In a recent study, we found that the cytochrome P450 system is used by cells to detoxify chemotherapeutic agents including MP-targeted lysosomes. When NADPH donor electrons are transferred to oxygen molecules by P450 enzymes as heme-containing monooxygenases, O^2−^ and H_2_O_2_ are produced. It is worth mentioning that the P450 enzyme can be found in the lysosomes, which are used to metabolize lysosomal medicines in addition to the cytoplasm. This process may promote P450-driven lysosomal ROS production, which, in turn, activates the lysosomal NOX2 (NADPH oxidase 2) system, thereby increasing ROS production. Therefore, the pH of the lysosome rises and Ca^2+^ is released (Figure 2) [54]. As a result of this molecular transformation, T-MPs used for drug packaging have the capacity to polarize M2 macrophages into M1 macrophages, suggesting that the effect of T-MPs on macrophage phenotype may be more than a single M2-type polarization, but a complex mixed phenotype. Further studies must be performed to determine the precise mechanisms.

Surprisingly, our findings suggest that MPs also stimulate adaptive responses from the immune system, especially in T cells and dendritic cells [56]. Neoantigens derived from tumor lysates have replaced primitive tumor cells in the development of vaccine production because of the technological advancements. However, the expression of tumor antigens by dendritic cells is a critical aspect of vaccination. Although T cells, the ultimate tumor cell-destroying actor, do not directly phagocytose T-MPs, dendritic cells (DCs) are able to successfully take up particles produced by tumor cells. We discovered that T-MPs can activate the cGAS/STING pathway, which results in the activation of NF-kB and IRF3 as well as the secretion of type I interferon through the loading of mitochondrial DNA [60]. T-MPs can also activate the expression of co-stimulatory molecules on the DC surface such as CD80 and CD86. In contrast to soluble antigens, T-MPs can encapsulate a variety of tumor antigens, which can be efficiently absorbed by dendritic cells due to their granular structure and optimal particle size.

Although tumor-cell-derived exosomes have tumor antigens that are expressed in parental tumor cells, they do not appear to be suitable for in vivo immune initiation or tumor vaccine development. Exosomes produced by tumor cells may suppress the immune system in multiple ways according to a number of studies, raising questions regarding their potential utility as tumor vaccines. T-MPs can produce greater protective immunity than exosomes or tumor lysates, and T-MPs loaded onto DCs can increase CD8^+^ T cell infiltration and IFN production, thereby greatly slowing the growth of tumors. In conclusion, T-MPs are excellent candidates for the creation of novel preventive and therapeutic cancer vaccines, since they not only cover the spectrum of tumor antigens, but also carry potential intrinsic signals. Future modified T-MPs will serve as therapeutic and preventative vaccines when loaded with particular antigens.

We discovered that T-MPs can be ingested by DCs and enter lysosomes, causing a transient rise in pH. As a result, T-MPs trigger the release of calcium ions from the lysosome, encourage the dephosphorylation of transcription factor TFEB into the nucleus, up-regulate the expression of CD80 and CD86, and promote DC activation and maturation. These benefits are related to the processing and presentation of T-MPs carrying tumor antigens, avoiding excessive antigen degradation, and facilitating the formation of MHC antigen–peptide complexes. The DNA components loaded in T-MPs can activate the cGAS-STING signaling pathway and promote the secretion of IFN-α by DCs, thereby promoting the maturation of DCs. These findings elucidate the precise molecular mechanisms by which CD8^+^ T cells are effectively fed with tumor antigens from T-MPs by DCs. Surprisingly, T-MPs provide a solid theoretical basis for the development of efficient and unique tumor cell-free vaccines for clinical applications.

Although intravenous or subcutaneous injection is the most popular way to administer tumor vaccines, recent developments in mucosal immunity have opened a new window for investigating oral delivery methods for preventive and therapeutic tumor vaccines. Oral vaccinations have long been used to successfully prevent fatal viral and bacterial illnesses [67,68]. However, they seem to be effective only against mucosal pathogens, and clinical trials of oral cancer vaccines have not been conducted. Oral administration offers many benefits over systemic delivery methods including simplicity, safety, and the activation of systemic mucosal and immunological responses. By combining a gastric acid inhibitor with an oral T-MP vaccine, our study demonstrated the effective inhibition of melanoma and colon cancer cell growth in mice [55]. T cells and DC cells were also stimulated during this procedure. Upon successful entry into the small intestine, T-MPs are mostly absorbed by ileal epithelial cells, where they activate NOD2, downstream MAPK, and NF-kB and produce chemokines such as CCL2, thereby chemoattracting CD103^+^CD11c^+^ DCs. While DCs capture T-MPs and cross-present tumor antigens loaded on T-MPs to activate systemic anti-tumor immune defense, ileal epithelial cells have the ability to convey T-MPs to stroma cells. By activating the NOD2 signaling pathway in ileal epithelial cells, the oral T-MP vaccine elicits specific T-cell immune responses; this study elucidates the cellular and molecular mechanisms by which T-MPs activate adaptive immune processes. It also highlights the potential clinical value of T-MPs in the development of novel oral vaccines.

### 4.2. T-MPs Activate Innate Immunity

Surprisingly, in addition to adaptive immunity, we found that T-MPs activate innate immunity. According to previous research, a substantial percentage of the tumor cells in malignant pleural effusions were rapidly and almost completely cleared in patients after the injection of drug-loaded T-MPs [52]. How does this work? In-depth analysis showed that drug-loaded T-MPs can quickly recruit neutrophils into the malignant pleural cavity and that activated neutrophils can destroy and remove tumor cells.

What causes pleural effusion to disappear so quickly? In addition to the need to destroy tumor cells, repairing damaged blood vessels also takes some time. When neutrophils die, due to a long evolution, the DNA and histones in their nucleus are released and form a network-like complex [52]. Neutrophils play an important role in fighting against pathogenic invasion. Due to the high viscosity of DNA, harmful bacteria can become entangled in and killed by this network structure. Consequently, these structures are referred to as neutrophil extracellular traps (NETs). We discovered that NETs can be released in cancerous pleural effusion. These highly viscous NETs act as efficient biomaterials that adhere to the surface of broken blood vessels like a plaster, preventing the outflow of intravascular fluid and allowing fluid in the pleural cavity to drain through lymphatic vessels, resulting in the rapid resolution of malignant pleural effusion (MPE) [52]. Neutrophils also kill tumors via NETs. Thus, it is possible to use neutrophils to treat clinically resistant malignant pleural effusion. Tumor immunotherapy is an effective strategy for tumor control that efficiently treats malignant pleural effusion by mobilizing the maximum number of neutrophils in the body, highlighting the special features of drug-loaded vesicles in tumor immunotherapy. The most recent clinical trial showed promising results in the treatment of malignant ascites, indicating that drug-loaded T-MPs are universally effective in treating clinical malignant pleural effusion [52,69].

We discovered that drug-loaded T-MPs can simultaneously stimulate neutrophils and its chemotaxis to produce anticancer effects. When drug-loaded T-MPs reached the lumen of the bile ducts above the cholangiocarcinoma tumor, some neutrophils were aspirated. Further research revealed that uridine diphosphate glucose UDPG and complement fragment C5a loaded in T-MPs is necessary for primary neutrophil chemotaxis [70]. When neutrophils reach the site of the obstructed bile duct, they destroy the matrix surrounding the cholangiocarcinoma cells, exposing the tumor cells, thus allowing them to come into contact with the drug-loaded T-MPs. This work showed that drug-loaded T-MPs trigger an innate immune response, suggesting promising applications for tumor immunotherapy in the future (Figure 3).

MPs are produced and taken up by different cells. As described above, drug-loaded MPs can be taken up by DCs and macrophages, and can promote the antigen presentation of DCs, reverse the phenotype of macrophages, and recruit neutrophils. Most previous studies have addressed the proteins and nucleic acids of extracellular vesicles. Interestingly, a new study on EV surface glycans, which have an important impact on EV heterogeneity and function, showed that in different cells, EV subpopulations of surface glycans were characterized differently. Modification of EV surface glycans revealed that the glycan form affected the EV cellular uptake and in vivo biodistribution [71]. This also provides some ideas for future research on the transformation and role of EVs.

## 5. Clinical Translation of EVs

The primary reason for death in cancer patients is tumor metastasis. Malignant pleural effusion (MPE) is widespread in many metastatic tumors and is untreatable [72]. Malignant tumor growth can disrupt the vascular structure of the pleural cavity during metastasis, resulting in a massive outflow of intravascular fluid and its retention in the cavity, leading to MPE [73]. The effusion is more pronounced if the tumor nodules obstruct the returning lymphatics. Additionally, because of the significant damage to the blood vessels, many red blood cells may pass through the capillaries and enter the pleural cavity, resulting in a hemorrhagic pleural effusion [74]. Many solid tumors including breast, ovarian, and lung cancer can spread to the pleural cavity and develop MPE [75]. For example, up to 60% of patients with advanced lung cancer develop malignant pleural effusions [76]. MPE is a serious threat to patient survival and quality of life, but there is no effective treatment in the clinical setting. Between 2015 and 2020, to treat MPE in non-squamous advanced non-small-cell lung cancer, we encapsulated T-MPs with methotrexate or normal saline in addition to a PP (pemetrexed/cisplatin) intrapleural perfusion regimen (ChiCTR-ICR-15006304). The methotrexate-loaded T-MP treatment group had a pleural effusion control remission rate of 82.50% among the 86 participants, which was substantially greater than the 58.97% rate in the control group. It was confirmed that drug-loaded T-MP technology improves the MPE treatment of patients with advanced lung cancer in a definitive, safe, and manageable manner.

MPs have been used in the clinical treatment of advanced malignancies with some success, but the sample size of these studies was not large and the route of administration was limited to local perfusion (chest, bile duct, bladder, etc.). The pharmacokinetics, safety, tolerability, patient benefit, and individual dosing regimens of MPs need to be explored in further clinical studies. In the meantime, the industrial preparation and quality control of large quantities of MPs remain a challenging issue. However, considering the many advantages and potential applications of MPs as described above, we are confident in the future of MPs.

With the continuing advancement of exosome research, some emergent exosome enterprises have made progress and conducted a series of clinical trials. Codiak, a leader in the field of engineered exosome (EV) drug delivery, employs its proprietary engEx platform to combine various drugs with well-designed engineered exosomes, targeting those exosomes to specific cells and tissues, altering the biological function of the recipient cells and producing the desired therapeutic effect [77].

Codiak has two anticancer medicines: an exo stimulator of interferon genes (exoSTING) and exoIL-12. exoSTING is designed for solid tumors and activates the “STING” receptor in immune cells. Patients with advanced/metastatic, recurrent, injectable solid tumors are the focus of phase I/II clinical trials designed to examine the safety, tolerability, pharmacological activity, and tumor response, with a focus on head and neck squamous cell carcinoma (HNSCC), triple-negative breast cancer (TNBC), anaplastic thyroid cancer (ATC), and cutaneous squamous cell carcinoma (cSCC) [78].

IL-12 does not fully operate as a cancer therapeutic, as many testing methods have shown it to be subject to unwanted systemic exposure and unpredictable pharmacological effects [79]. In order to overcome these drawbacks, Codiak created exoIL-12, a uniquely designed exosome therapy that exhibits functional IL-12 on the surface of the exosomes. By fusing IL-12 to the common exosome surface protein PTGFRN, a similar potency recombinant IL-12 (rIL-12) was produced in vitro. After intramural injection, exoIL-12 showed a longer tumor retention period and more potent anticancer effects. The N-terminal coding region of the exosomal membrane protein PTGFRN was fused to the IL-12 subunits p35 and p40 to generate the exoIL-12 candidate protein, which was then stabilized by a flexible linker. Purified designer exosomes essentially free of protein and nucleic acid contamination were isolated from cloned cell culture supernatants using a discontinuous density gradient. A series of in vitro investigations validated the functional display of IL12 and showed equivalent activity in the T-cell and NK cell activation assays.

In 2020, Codiak announced that the preliminary stage of their Phase I trial testing a single ascending dosage of exoIL-12 in healthy volunteers had achieved its main objective. In this randomized placebo-controlled double-blind study, ExoIL-12 showed satisfactory safety and tolerability with no measurable systemic exposure to IL-12 and no adverse treatment-related local or systemic events [80].

Carmine Therapeutics has created a new gene treatment that employs EVs derived from red blood cells (RBCs). The Carmine Therapeutics REGENT^TM^ platform uses electroporation to insert therapeutic genes into RBCEVs, which offers the benefits of numerous loading fragments, multiple types, and positive outcomes [81].

In September 2019, Alligator began preclinical testing of its exosomal medicine 4224 in Sweden. The bispecific antibody 4224 activates T lymphocytes to attack tumor cells, targeting CD40 and EpCAM in tumor exosomes. It is currently in the drug development phase, which entails locating novel drug action mechanisms, creating and refining new drug candidates, assessing preclinical efficacy and safety, and ultimately conducting confirmatory clinical studies in cancer patients.

Extracellular vesicles (including exosomes), released from bone marrow mesenchymal stem cells have been generated, isolated, and purified by Aegle Therapeutics (MSCs). The FDA authorized the company’s novel EV medication AGLE-102′s Investigational New Drug Applications (IND) in April 2018 and May 2019, and a Phase I/II clinical trial for the treatment of dystrophic epidermolysis bullosa (DEB) began in 2021 [81].

Avalon GloboCare focuses on the development of engineered exosomes generated from mesenchymal stem cells. The main products are AVA-201, AVA-202, and AVA-203, which promote angiogenesis and wound repair for the treatment of liver fibrosis and pulmonary fibrosis, respectively. A joint laboratory has been established with the Shanghai Ninth People’s Hospital to conduct research on regenerative exosomes to accelerate the development of diagnostic and therapeutic applications.

Vesigen Therapeutics uses a fusogenic extracellular vesicle delivery technology to advance groundbreaking therapies targeting intracellular targets. Their patented technology, called ARMMs (ARRDC1 Mediated Microvesicles), enables the delivery of a wide range of payloads including RNAs (mRNA, shRNA, ribozymes), proteins (signaling proteins, enzymes, antibody fragments), and gene-editing complexes (Cas9/gRNA) directly into the cytoplasm of target cells, expanding the range of targets [82].

Fc-EVs are antibody-displaying extracellular vesicles developed by Evox to deliver tumor-targeted therapy. In experiments with melanoma tumor-bearing mice, Fc-EVs, Fc-EVs + DOX, Fc-EVs + PDL1 ab, and doxorubicin-loaded intraperitoneal injection treated malignant melanoma and prolonged survival in mice compared to Fc-EVs alone.

While basic exosome research is currently accelerating, the industry is in its infancy and the competitive landscape is yet to be shaped. Exosomes have been known for nearly 40 years, but it was not until the concept of “precision exosomes” was introduced in 2015 that exosomes became a research hotspot for disease diagnosis and precision therapy, with great promise for application [83].

Although MPs have been widely used in clinical practice, there are still limitations. These include limitations as follows. (1) T377he preparation method: The current main process for the preparation of MPs of tumor cell origin is the production of tumor cells with apoptosis induced by UV radiation. The intensity and duration of UV irradiation is a key factor affecting the production of MPs. If the irradiation time is too short and or the UV intensity is too low, the apoptosis of tumor cells will be reduced and the release of MPs will be decreased; if the irradiation time is too long or the UV radiation is too strong, it may lead to cell rupture and death, and the obtained MP film will be incomplete. (2) Separation techniques: currently, our laboratory isolates MPs of tumor cell origin, mainly using differential centrifugation. The supernatant of the UV radiation-treated cell culture is collected and subjected to differential centrifugation to obtain the final product. In this process, some exosomes with a large particle size may not be effectively removed by differential centrifugation. In addition, excessive centrifugal force may have some effect on the membrane integrity of MPs. (3) Timeliness: Drug-carrying vesicles are stable when stored at 4 degrees for 7 days, but may gradually decrease in efficacy as the storage time increases.

## 6. Conclusions

This review outlines the methods of vesicle isolation, clinical applications, and translation. The innovation of this review is to summarize the research progress on vesicle immunotherapy based on the research content of our study. Tumor immunotherapy is at a watershed moment on the path to humanity’s eventual victory over cancer. PD-1 antibodies and CAR-T are representative of tumor immunotherapy, and have been listed as one of the top 10 technological advances in 2013 in *Science* The Nobel Prize of Physiology and Medicine in 2018 was awarded to Tasuku Honjo and Allison James for their contribution to immune checkpoint tumor therapy. However, tumor immunotherapy still has many obstacles to overcome. In terms of clinical treatment, the targeted population is small and the therapeutic effect is unsatisfactory. In terms of scientific theory, immunotherapy currently only targets immune cells and lacks new methods that simultaneously target immune and tumor cells. In terms of national strategy, original therapies in China are difficult to achieve. Further research on tumor immunotherapy still needs to be conducted, both scientifically and creatively. In addition to the ability to target and eliminate tumor cells, particularly tumor stem cells, drug-loaded T-MPs play a significant anti-tumor function through the activation of innate immunity and adaptive immunity. Drug-loaded T-MPs have great potential for engineering as well as medication delivery. In this review, we examined the tumor-cell-derived MPs and even TRC-derived MPs, which have low heterogeneity and high safety because they are natural drug carriers, as drug delivery vehicles for cancer therapy. In addition, TRC-derived MPs, because of their greater dryness and softer surface, can better attack TRC and reverse the drug resistance of tumor cells. Meanwhile, normal cells can also produce MPs; our study (unpublished) demonstrated that erythrocyte-derived MPs have good therapeutic effect in the treatment of tumors because extracellular vesicles derived from erythrocytes do not contain nuclear DNA and mitochondrial DNA and therefore do not involve risk to the recipient genome. Red blood cells are the most numerous cells in the human body and are the most convenient cells to realize the preparation of clinical-grade extracellular vesicles. Allogeneic blood transfusion is safe and extracellular vesicles derived from red blood cells are natural ingredients, so they have high biocompatibility and can avoid self-limitation. Currently, regulatory authorities have given the go-ahead for the use of drug-loaded T-MP tumor therapy technology for advanced clinical malignancies in several provinces and cities. This technology is anticipated to significantly raise the quality of life for patients with malignant tumors and lessen their financial burden. However, there is a need for an integrated system for the preparation, characterization, mass production, and quality control of exosomes.

## Figures and Tables

**Figure 3 bioengineering-10-00325-f003:**
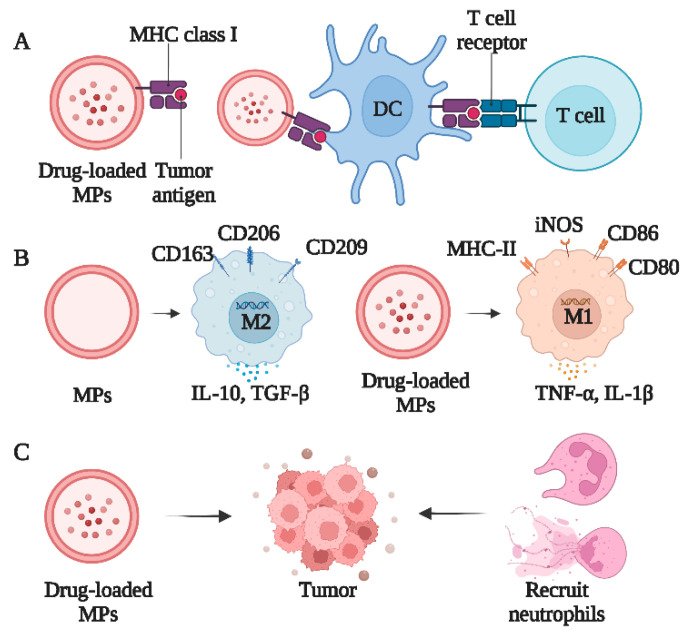
The dual significance of T-MPs in immune system modulation. (**A**) Drug-derived MP facilitate DC antigen presentation based on [58] (Cancer Immunology Research, 2018); (**B**) drug-derived MP promotes the M1 phenotype of macrophages based on [60,66] (OncoImmunology, 2016; Cellular & Molecular Immunology, 2019); (**C**) drug-derived MP recruits neutrophils based on [52] (Cancer Immunology Research, 2020).

## Data Availability

Not applicable.

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
