# Peer review of "Challenges and Opportunities for Extracellular Vesicles in Clinical Oncology Therapy"

_bioengineering, 2023, doi:10.3390/bioengineering10030325_

Round 1
Reviewer 1 Report
The manuscript “Challenges and Opportunities of Extracellular Vesicles in Clinical Oncology Therapy" submitted by Tang Ke is up to date. This research helps to better understand the role of extracellular vesicles in cancerogenesis, immunity and also in others physiological processes in the body.
I have several comments that should be corrected:
I miss table with abbreviations1. miss table with abbreviations. The MPs you have not explain when you use it first time,
2. For the part 4.1 I suggest you to prepare some simple scheme/picture for better understanding.
After incorporation of all of these comments I suggest to publish this experimental manuscript.

Reviewer 2 Report
The manuscript of Shuya et al. illustrates the potential use of Tumor Microparticles (T-Mps) for developing an immunotherapeutic application in cancer management. The review recapitulates the scientific work done by this Research Group, for which the relevant findings have been clearly presented with high relevance in this field. The paper is timely with respect to the state-of-the-art in the field and demonstrates a broad background, with deep knowledge in the area by the Authors and provides a good outlook on the development of this application (with a specific focus of China research in the field). The acronym MPs was used before it the first time is introduced. The description of EVs' isolation methods is not entirely justified and adequately connected with the other paragraphs (only partially when extracellular vesicles are introduced). The contents of the review are original (except for a single study from 2022, DOI: 10.1002/adma.202201054), the quality of the discussion and critical view about the regulatory, technical, and methodological limitation that hinders the development of new therapeutic tools based on T-MPs is insufficiently addressed. A few aspects, such as fate, biodistribution (tracking in vivo the whole body biodistribution) and inter-organ interplay, are not adequately presented and discussed. Also, the quality of the presentation of the ongoing clinical trials is oversimplified and descriptive, without deep analysis and discussion of barriers being detected and of the challenges for the clinical translation of these new treatment options being tackled by the researchers.
Reviewer 3 Report
Excellent, well written, paper. Given novelty of opportunities of application of exosomes in clinical oncology, I would suggest to add subparagraph on "Research Directions and Potential Future Applications of Extracellular Vesicles in Oncology" and perhaps some hypothetical at this point applications of EV.
Speaking about the future: authors focus is on "delivery of agents" but, for example, could exosomes be utilized to deliver L-methioninase to induce methionine starvation of cancer cells ? In other words instead of delivery they would be used for subtraction of essential for cancer cells nutrients.
Reviewer 4 Report
This is a very comprehensive and informative review of the diverse applications of tumor cell-derived EVs. However, major EVs that are currently under development or clinical trials are those derived from normal cells or mesenchymal stem cells. In this regard, the reviewer would like to suggest the authors to discuss the safety issues and advantages/disadvantages of EVs, especially tumor cell-derived EVs versus normal cell- or stem cell-derived EVs.
Round 2
Reviewer 2 Report
The Authors have sufficiently addressed the Reviewer's concerns and added comments and additional references rendering the manuscript suitable for publication.
